# 6G to Take the Digital Divide by Storm: Key Technologies and Trends to Bridge the Gap

**Chiara Suraci [1], Sara Pizzi [1,\*], Federico Montori [2], Marco Di Felice [2] and Giuseppe Araniti [1]**

[1] Department of Information Engineering, Infrastructure and Sustainable Energy (DIIES), University Mediterranea of Reggio Calabria, 89100 Reggio Calabria, Italy; chiara.suraci@unirc.it (C.S.); araniti@unirc.it (G.A.)

[2] Department of Computer Science and Engineering (DISI), University of Bologna, 40127 Bologna, Italy; federico.montori2@unibo.it (F.M.); marco.difelice3@unibo.it (M.D.F.)

[\*] Correspondence: sara.pizzi@unirc.it

**Abstract:** The pandemic caused by COVID-19 has shed light on the urgency of bridging the digital divide to guarantee equity in the fruition of different services by all citizens. The inability to access the digital world may be due to a lack of network infrastructure, which we refer to as *service-delivery divide*, or to the physical conditions, handicaps, age, or digital illiteracy of the citizens, that is mentioned as *service-fruition divide*. In this paper, we discuss the way how future sixth-generation (6G) systems can remedy actual limitations in the realization of a truly digital world. Hence, we introduce the key technologies for bridging the digital gap and show how they can work in two use cases of particular importance, namely eHealth and education, where digital inequalities have been dramatically augmented by the pandemic. Finally, considerations about the socio-economical impacts of future 6G solutions are drawn.

**Keywords:** 6G; digital divide; NTNs; AI; XR; MEC; affective computing; BCI; D2D; use cases

## 1. Introduction

The digital divide has existed since access to the Internet began to spread among the worldwide population. This phenomenon consists of the gap between people who have access to the digital world and those who have not, for various reasons, such as geographical location, economic status, level of education, and general interests. Nonetheless, the health emergency triggered by the COVID-19 propagation has opened our eyes to the difficulties related to a life far from the digital world. Among others, Nokia collects some statistics relating to the inclusivity in distance teaching activities, undertaken during the lockdowns caused by the pandemic, pointing out that, according to UNICEF, 31% of the school children in the world were unable to access remote learning [1]. From a different perspective, Ericsson provided some data on the impact of the pandemic in the U.S. wireless communications industry, reporting a 19.6% increase in data traffic, 24.3% in voice traffic, and 25% in texting; moreover, the authors state that the COVID-19 spread has highlighted the true face of the digital gap, which is not only an access problem but, more generally, it is caused by lacks in affordability, quality of coverage, and technical skills [2]. This thesis is supported even by Huawei which emphasizes the fact that 50% of our planet has no Internet access and, thus, presents the TECH4ALL project, aimed at expanding the granting of digital rights by acting on three core fronts: technologies, applications, and skills [3]. The Cisco Annual Internet Report claims that about 2/3 of the worldwide population will have access to the Internet by 2023, with an estimate of almost four devices per capita [4]. Bridging the gap must not be considered only a cost: the forecasts from Vodafone in [5] mention that the cumulative additional contribution to the GDP of new digital technologies could amount to 2.2 trillion euros in the EU by 2030. Furthermore, this report highlights the benefits obtainable through the application of a digital-by-design

approach in the recovery plan for Europe following the COVID-19 crisis: enhanced quality of life for citizens, long-term economic growth, lower resource consumption, increased resilience and fairness of society.

Statistics and forecasts are shown in Figure 1 prove that the digital divide problem is very striking and multi-dimensional. Indeed, the likelihood that people have access to the Internet is influenced by the level of economic and cultural development of the country in which they live; besides, the generation gap impacts the digital divide; even the gender is a factor of diversity when looking at the number of people open to digitization. Whatever the reason that causes the impossibility or unwillingness of some people to *live connected*, today, the digital divide represents a real obstacle to the recovery that the world needs following the COVID-19 pandemic, which has befallen the global population some time ago and which is still conditioning our lives. Although some countries were more digitized than others, similar disparities in Internet access for households with a higher level of poverty and rural areas were reported worldwide and even before the pandemic began [6]. This led us to consider the reassessment of numerous aspects concerning the enforcement of information and communications technologies (ICT) in various areas of society, including work, education, and health. Unfortunately, none of the existing solutions has been proved to be an effective response to the digital divide problem, being too focused on specific demand or introducing high costs for its development. For this reason, in this paper, we investigate how the problem of the digital divide can be mitigated by looking to the future, specifically to the sixth generation (6G) and rising technologies, being the latter not just a mere exploration of more spectrum at high-frequency bands [7] rather a new paradigm for ubiquitous, pervasive and high-speed Internet connectivity. Furthermore, 6G could represent the turning point as it will allow the achievement of a high level of automation in the execution of various services and the extension of coverage of cellular networks. This is the reason why this work refers to 6G technologies, classified as *evolutionary* and *revolutionary*, wherein the former has already emerged with the fifth generation (5G), but is not yet widespread on the market [8]. In more detail, we provide three main contributions to this study. First, in Section 2 we discuss the evolution of the digital divide concept and the challenges that can be addressed by the technological development; differently from [9], which focuses on the specific issue of coverage of remote areas, our study proposes a multi-dimensional discussion, by further distinguishing between *service-delivery divide*, from a network-oriented perspective, and *service-fruition divide*, from an individual-oriented perspective. Second, in Sections 3 and 4, we discuss how the future 6G network is expected to overcome both the issues, by identifying trending technologies that should be further developed and be part of the upcoming specifications. The role of Artificial Intelligence (AI) and big-data collection and analytics via Machine Learning (ML) techniques is transversal to service delivery and fruition and for this reason is discussed apart in Section 5. Third, in Section 6, we describe selected use cases (e.g., eHealth and education) where digital inequalities have been dramatically augmented by the COVID-19 pandemic, and how the aforementioned 6G technologies could be effective in bridging the gaps. Considerations about the socio-economical impacts of future 6G solutions are drawn in Section 7.

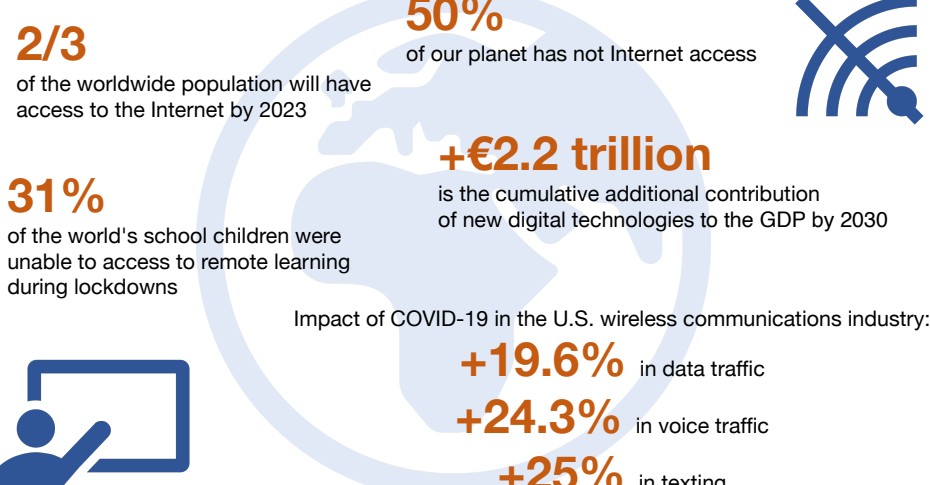

**Figure 1.** Some numbers on the Digital Divide.

## 2. The Evolution of the Digital Divide

The origin of the digital divide dates back to the mid-nineties when, in the U.S., through the publication of some reports, the differences between people with access to the Internet ("haves") and those without ("have-nots") began to be analyzed. The phenomenon of the digital divide has evolved over time, passing from the *first-level*, related to the problems of access and connectivity, to the *second-level*, consisting in the lack of the necessary skills to properly exploit ICT, and finally to the *third-level*, concerning the differences in the outcomes and consequences obtained by using the Internet [10].

According to this, we classify the different types of digital divide into two major categories: the *service-delivery divide* and the *service-fruition divide*. The former concerns the digital exclusion caused by the absence of network infrastructures necessary for Internet access and digital services delivery; the latter can be considered related to a person-specific divide, since an insufficient level of digital literacy or a set of physical inabilities could prevent people from enjoying the benefits deriving from the fruition of digital services.

The existence of these conditions not only affects the origin and the past of the digital divide phenomenon, but also concerns the current time. As a matter of fact, the state of health emergency we are still experiencing has exacerbated the digital gap. In the enterprise context, the resilience of companies has been enabled only for those that invested in technological innovation. This has represented a lifeline for the operational continuity of the businesses that had the readiness to carry out the *digital metamorphosis* path, necessary for survival in the period of COVID-19. Similarly, receiving the provision of numerous services in telematic and innovative modalities has proved to be straightforward only for the part of the population inclined to use ICT; all the others encountered not a few difficulties in adapting to the new set-up imposed by the measures implemented to limit the pandemic. Even in light of the reports cited in Section 1, we can state that the digital exclusion corresponds to exclusion from access to services. Indeed, going through the pandemic caused by COVID-19, we have realized how important technology is as a means of keeping people in contact with the outside world by digitally receiving different types of services. This is the reason why we define a service-based classification to group the causes of the digital divide.

Regarding the future evolution of the digital gap phenomenon, *normalization* and *stratification* are the two contradictory predictions that have been defined in the literature [11]. According to the first, over time the differences that cause the gap will gradually disappear until they reach saturation; it relies on the belief that government institutions will succeed in promoting and facilitating Internet access in the long run. Conversely, the second promotes the idea that the digital divide will unavoidably grow, owing to a continuing

tightening of disparities within societies. This perspective appears to be more realistic, since some opinions exist according to which the evolution of mobile wireless networks could worsen the digital divide. For example, the very fact that, in this first phase of release, the 5G standard has been unevenly distributed in different countries of the world, led to the establishment of a further gap between areas that receive state-of-the-art network coverage and those that don't. Again in [11], it is stated that *connectivity does not end the digital divide, skills do* to emphasize that, anyway, increasing network coverage alone may not suffice to bridge the digital divide, instead, technology ought to be exploited to bring people closer to the digital world.

The described evolution path is represented in Figure 2.

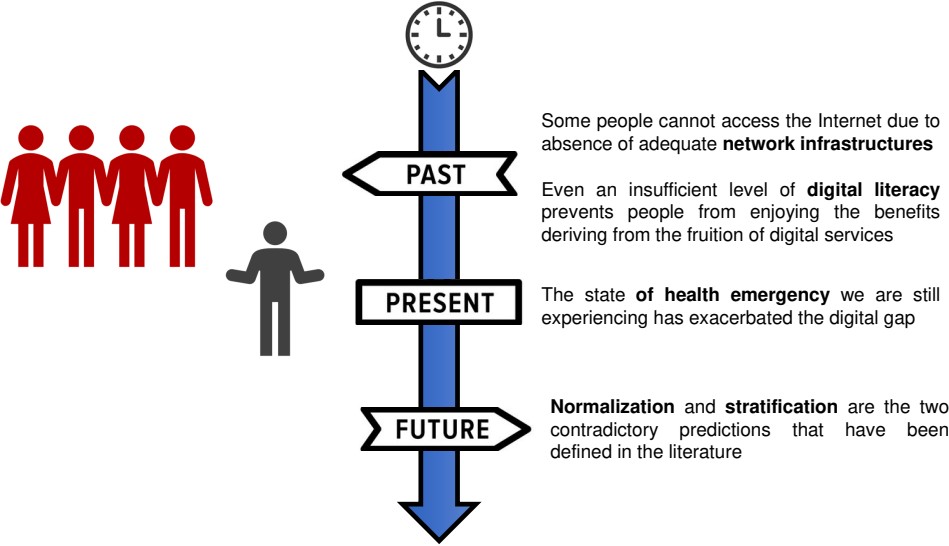

**Figure 2.** The evolution of the Digital Divide.

## 3. The Technologies for Bridging the Service-Delivery Divide

In this section, we examine the possible implementations of technologies that can be considered powerful tools to face and govern the problem of the service-delivery divide, as they would be able to offer Internet access even where traditional network infrastructures fall short. Some of the technologies that will be mentioned belong to the evolutionary category (see Section 1) as they have been already considered promising for the deployment of 5G but they are not yet suitably widespread.

### 3.1. Non-Terrestrial Networks

Service continuity and scalability are two prerogatives of ubiquitous connectivity that can be fostered, in 5G and beyond systems, with the integration of terrestrial and non-terrestrial networks (NTN) [12]. NTNs encompass spaceborne and airborne systems. Satellites may be either geostationary (i.e., GEO) or orbit around the Earth at low or medium orbits (i.e., LEO and MEO). Among LEO satellites, a growing interest has risen towards CubeSats, a new class of miniaturized satellites known for their very small dimension and low cost, hence enabling their deployment in mega-constellations to provide global connectivity and large throughput. Equally interesting are Unmanned Aerial Vehicles (UAVs) that may be deployed in swarms to provide an on-demand aerial infrastructure where needed [13]. Based on these features, NTNs can come into play to mitigate the digital gap, by ensuring connectivity to a massive number of Internet of Everything (IoE) devices where terrestrial networks can fail, therefore in disadvantaged areas, in emergency scenarios, and highly crowded environments.

### 3.2. Exploiting Higher Frequencies

To truly bridge the digital divide, ultra-high-speed communications required to deliver 6G services need to be enabled. A very promising solution in this direction is to exploit new portions of the spectrum [14]. While the millimeter-wave band has been already considered in 5G mobile networks, now the interest is moving towards TeraHertz and even optical bands, motivated by the very recent advances in electronics and photonics that enabled the manufacturing of portable equipment operating at such frequencies. However, signal propagation becomes critical due to severe path loss, high molecular absorption, and the requirement for very precise antenna pointing. Coverage extension may be achieved by means of Intelligent Reflecting Surfaces (IRS) [15] that allow us to realize a virtual line-of-sight (LoS) link by smartly reconfiguring the wireless propagation environment [16]. Mainly, IRSs exploit massive low-cost passive reflecting elements integrated on a planar surface that independently reflect the incident signal by controlling its amplitude and/or phase and thus collaboratively achieve fine-grained three-dimensional (3D) passive beamforming for directional signal enhancement. The service-delivery divide could benefit from the use of higher frequencies in application scenarios that span from indoor coverage to Earth-to-ground communications.

### 3.3. Device-to-Device

Although technologies such as Software-Defined Networking (SDN) have emerged as enablers in the evolution process of 5G networks [17], the trend to evaluate distributed networking approaches to extend the network coverage and scalability has gained great momentum. On the eve of 6G, this still represents a key solution to boost the access to connectivity, therefore, also bridging the digital divide. Particularly, Device-to-Device (D2D) communications could be harnessed to master the problem of the limited distance that the higher frequency waves exploited in future 6G networks can cover [18]. Actually, relay nodes could be exploited to forward the signal through the establishment of direct communications, thus extending network coverage and facilitating access to services even to devices outside the antenna's LoS. D2D communications are characterized by high speed and low latency thanks to the proximity between end-devices [19], hence they are entirely in line with the requirements of the upcoming 6G networks.

### 3.4. Multi-Access Edge Computing

In the vein of the previous discussion on the shifting to distributed networking paradigm, also the Multi-access Edge Computing (MEC) is increasingly catching the eye, since it allows to improve the delivery of services in several respects. First of all, the resources of MEC servers can be provided, through virtualization, to limited-resources consumers based on the most appropriate service model (Infrastructure as a Service - IaaS, Platform as a Service—PaaS, Software as a Service—SaaS) [20]. Then, the proximity of the MEC to users enables various benefits, including low latency and context-awareness, which allows us to customize the service delivery to the needs of the consumers. The authors of [21] cite the efficacy of MEC in providing support to communication, computing, and storage, thus improving the Quality of Service (QoS) provided to users. This can be considered a plus in bridging the digital gap as poor QoS is seen as an impediment to the effective delivery of bandwidth-intensive services.

## 4. The Technologies for Bridging the Service-Fruition Divide

Over the past couple of years, the presence of a pandemic has accentuated the problem of the service-fruition divide, affecting those who have inabilities to access digital services, not because of infrastructural issues, but rather of their physical conditions, handicaps, age, or digital illiteracy. Applications revolving around 6G are proposing to support a set of digital interfaces at a large scale, not only in localized and specialized environments. This is possible for the first time in history, as these interfaces, while promoting ease of use and human centrality, have requirements in terms of latency and reliability that are too

stringent to overcome with legacy technologies. The most debated solutions are presented in this section.

### 4.1. Extended Reality

Extended Reality (XR), which encompasses key terms such as Virtual Reality (VR), Mixed Reality (MR), and Augmented Reality (AR), is believed to drive several killer applications in the 6G era. 5G requirements fall short in supporting such applications, which need ultra-reliable low latency communications and cannot rely on mobile broadband [7]. In particular, XR pervasive applications will depend not only on the networking constraints, but also on the perceptual and sensory ones, which should be aligned with the above (i.e., tolerate delays that are imperceptible to human senses). The data rate is one of the major obstacles as, even now, many of the existing XR applications struggle in moving from wired to wireless, in particular for WAVAR (Wide Area VR, AR, MR) which aims to provide ubiquitous wireless services for XR [22]. WAVAR demand 6G technologies for ultimate application—as opposed to their local counterpart, LAVAR—in terms of data rate requirements (1Tbps peak rate), which cannot be satisfied by current 5G deployments [23]. If requirements are met, even digital illiterates would be able to interact with a controlled digital world through actions and perceptions that leverage all five senses (haptic, gestures, sound and speech, virtual sight, etc.). Over the past years, XR has been proposed in a variety of use cases connected to teaching activities [24] and aiding people with disorders [25], however, these use cases are typically experimental or extremely localized. It is evident how the potential of overcoming the service-fruition divide may dramatically increase if these technologies become pervasive. Challenges rise now in designing unique metrics capable of capturing both network requirements and the physical experience of users.

### 4.2. Brain-Computer Interfaces

Brain-Computer Interfaces (BCI) have been used extensively in assisting elderly or disabled people. While wired BCIs have been active for years, wireless BCIs are less supported due to their stringent QoS and Quality of Experience (QoE) requirements [26]. Over the last years, we can observe a handful of wireless BCIs implemented through short-range communication technologies in LANs, for use cases like home automation and digital healthcare [27]. However, these technologies are far from bridging the service-fruition divide, as their pervasiveness is limited to localized areas and hardcoded functions (e.g., switching on and off smart bulbs, etc.). With the perspective of 6G, requirements for deploying BCIs at large could be met and the deployment of a set of compelling applications in, for instance, urban environments could become a reality. Just like (and probably more than) XR, BCIs may be the new frontier of Human-Computer Interaction, involving Internet of Things (IoT) devices that will pervade our urban realities and will enable 6G ultra-low-latency connections. This also involves applications in healthcare that have, so far, only been conducted in controlled environments [28]. Such a giant leap in overcoming the service-fruition divide is envisioned to mark the end of the smartphone era, in a decade from now. Using wireless Brain-Computer Interaction technologies instead of smartphones, people will interact with their environment and other people using discrete devices, some worn, some implanted, and some embedded in the world around them.

### 4.3. Affective Computing

Affective computing encompasses a set of use cases that will be particularly enhanced by 6G technologies. It refers primarily to devices that can adapt their service provisioning schemes according to the mood and the emotions of the final user [29]. The first studies on the concept were presented a couple of decades ago, while recently it experienced a revamp due to its natural applicability on smartphones [30]. Now, 6G brings a set of concepts to the table that will change the way in which service provisioning takes place. One such is Human-Centric Services (HCS), a set of services that put the final user in the foreground and match her or his requirements to network performances, making

affective computing potentially more pervasive than ever [7]. This is specially tailored to the educational use case, for instance in enhancing the relationship between teacher and student in online classes through online automatic learning processes fed by physical parameters (e.g., posture, speech, and expression) [31]. This results in involving more individuals who would instead be cut out due to e.g., lack of attention.

Table 1 shows a classification based on the possible employment of each technology described in Sections 3 and 4 for the resolution of the major categories of digital divide (i.e., service-delivery and service-fruition).

**Table 1.** Classification of technologies based on the mastered type of Digital Divide.

| Technology | Service-Delivery | Service-Fruition |
| --- | --- | --- |
| Affective Computing | ✗ | ✓ |
| Artificial Intelligence (AI) | ✓ | ✓ |
| Brain-Computer Interface (BCI) | ✗ | ✓ |
| Device-to-Device (D2D) | ✓ | ✗ |
| Exploiting Higher Frequencies | ✓ | ✗ |
| Extended Reality (XR) | ✗ | ✓ |
| Multi-access Edge Computing (MEC) | ✓ | ✗ |
| Non-Terrestrial Networks (NTN) | ✓ | ✗ |

## 5. Artificial Intelligence: The Ultimate Breakthrough?

AI plays a role of paramount importance in the evolution process that wireless networks are experiencing. Its use can enhance the performance of many applications by providing a wide range of beneficial properties. In particular, ML is a branch of AI that is considered a top solution in many tricky 6G applications [32]. The ML technology allows us to train systems that, by the processing of collected data, can learn patterns by experience and, consequently, improve the performance and quality of the offered services. In particular, following the recent progress in deep learning as well as the advent of smart devices that are capable of processing ML algorithms on the edge, the wireless community has gained a renewed and huge interest in such technologies, which can now be leveraged in use cases that were unable to support them before. Now, with edge AI and ML we can envision networks of heterogeneous objects that are self-organizing and can meet high KPIs even in harsh scenarios via, e.g., reinforcement learning [7]. In this context, we are witnessing the shift of AI and ML components closer and closer to the edge, to the point that ML is projected to be an actual part of 6G technologies, rather than something that builds on top of them as it was with previous generations. Another remark is that this transition is expected to take place transversely, which means that potentially all 6G technologies will be affected at once.

This section aims to bring forth the potential achievable by applying AI to the purpose of lowering the gap brought by the digital divide. Current trends suggest that both the service-delivery gap and the service-fruition gap would greatly benefit from embedding AI into shared resources. In such a context, any single node of the network will produce data about connectivity, environment, and such, and the collection of big data from IoT scenarios is a key enabler to better understand the challenges of various nature in Internet access. Specifically, data is then analyzed through ML to create more inclusive and scalable networks. For example, data could be gathered and investigated to comprehend which categories of people make better use of the benefits provided by ICT and which ones find difficulties in doing so. In [10], a predictive ML technique is implemented to analyze the primary socioeconomic factors that cause the digital gap in Spain. According to [33], ML can improve the network performance through the undertaking of *adaptive network optimization actions*, enabled by the ability to learn from the wireless environment that ML

provides to the network infrastructure. In such a sense, there are a lot of potential usages that meet the purpose of bridging the digital divide. Certain types of network traffic, if their nature is well understood by an AI engine, could be privileged (e.g., remote health diagnosis, online lectures) in contrast to others (e.g., entertaining). Historical network parameters observed by edge nodes can also feed a fog/edge ML model so that the distribution of network resources could be locally automated. Moreover, *planning* capabilities could be introduced to lower the risks of shortages under normal resource usage. This is also crucial to coordinate with mobile and on-demand network resources, as in NTNs, forming a real "collective network intelligence" [7] to overcome the service-delivery gap and bring resources where and when they are most needed. On the other hand, AI is also a powerful tool for improving the accessibility of innovative digital services for people with a low level of digital literacy, notably the elderly, people with disabilities, and people living in underdeveloped areas. Usages in this direction are often mentioned in the literature, for example, in [34], where authors survey some works concerning the application of technology to the provision of accessible cultural heritage sites experiences, also highlighting the importance of the role of AI in adapting the offered experiences to the target audience. Moreover, most of the technologies that we presented for bridging the service-fruition gap rely on AI and ML as their core enablers. This entails that AI could be a powerful tool for overcoming the digital cultural and cognitive gap, as it could be used to support those who would otherwise remain "digitally excluded".

In Table 2 we match the major AI trends to the 6G driving technologies to show that AI is by now an orthogonal trend that embraces all of them in different guises.

**Table 2.** AI candidates for the 6G technologies.

| 6G Technology | AI Candidates |
|---|---|
| Sensor Networks and Edge Systems | Reinforcement Learning, Unimodal & Multimodal Classification, Autoregressive Models |
| SDN & D2D | (Deep) Reinforcement Learning, Multimodal Classification, Deep Neural Networks |
| Drone Swarms | Reinforcement Learning, Unsupervised Learning, Agent-based models |
| Extended Reality | Computer Vision, Autoregressive models |
| Brain-Computer Interfaces | Recurrent Neural Networks, Autoregressive Models, Continual Learning |
| Affective Computing | Convolutional Neural Networks, Multimodal Classification, Natural Language Processing |

## 6. The 6G Services to "Connect" People

In this section, we describe two use cases for which leaps and bounds need to be made in the area of digitization: eHealth and education. The main weaknesses of the current digitization status of these services are highlighted, to demonstrate that, in these fields, existing access difficulties for some categories may be overcome by the aforementioned 6G technologies.

### 6.1. The eHealth Case

6G will mark a turning point in the digitization process of the healthcare sector and the application of new paradigms can help in achieving a *fully digital and connected world* [35]. This represents an important step forward in overcoming the problem of the digital divide. Thanks to the provision of telemedicine services, the physical barriers of separation between patients and health professionals can be surpassed and the delivery of health services can be

remotely guaranteed to many people, wherever they are whatever their condition. Indeed, to integrate telemedicine into the health systems of the different countries of the world, training courses should be taught to medical personnel and inclusive solutions should be designed to allow access to digital services even for people culturally far away from the technological world.

Figure 3 depicts the eHealth scenario consisting of the caring @home of a patient residing in a rural/critical area. Internet access to the patient premises may be provided by means of either spaceborne or airborne vehicles to enable its health monitoring by means of Medical Things (IoMT) devices, possibly exploiting D2D communications among them to extend indoor coverage. Due to the likely impossibility/inability of the patient to interact with the eHealth devices, AI and BCI can come to the rescue for contriving the fittest remedies. For example, IoMT devices that can obey voice commands and execute precise instructions can be delivered to the patients to allow them to remotely manage simple monitoring operations of health parameters. Furthermore, MEC servers can be installed close to the IoMT devices clusters (i.e., on the UAVs) to lighten their workload through computation offloading and reduce the data-gathering delay.

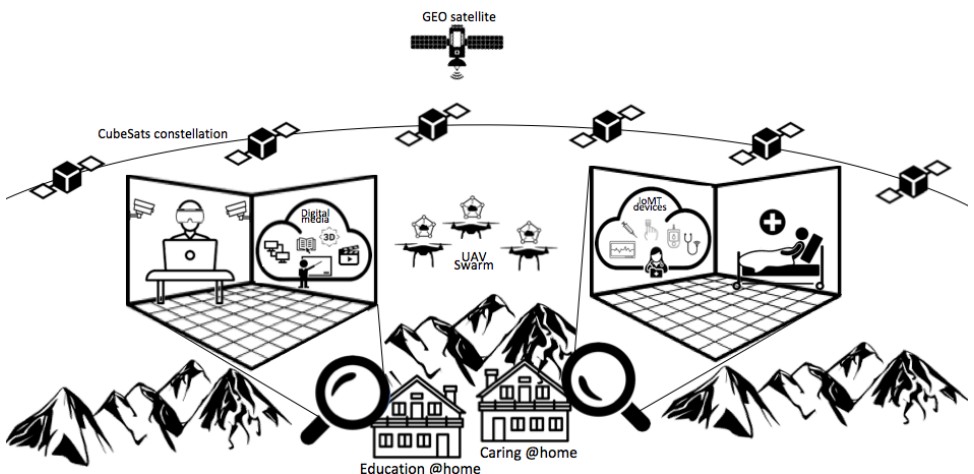

**Figure 3.** The analyzed use cases where 6G can bridge the digital divide: caring and education @home in rural areas.

Remote monitoring and assistance of patients are just some of the eHealth services that can be efficiently managed thanks to mobile wireless networks and that could benefit a lot from the advantages offered by 6G technologies. For example, specialist teleconsultation between doctors operating in different parts of the world is a very important application that would allow to break down the geographical boundaries of knowledge, making the experience and skills of medical luminaries available everywhere. Remote surgery, which can be defined as the telehealthcare service thanks to which a doctor can perform surgery at long distances from the patient [36], is an example of an application that asks for stringent requirements in terms of latency, security, and reliability in order to be widely diffused and that, therefore, would greatly enjoy the support of future 6G networks. A different use of technology in support of healthcare is shown in [37], where the benefits of applying VR for motor rehabilitation are deeply described. There are also numerous research proposals that introduce the use of potentially key technologies of 6G for eHealth services related to COVID-19, such as [38–40].

### 6.2. Education

The COVID-19 pandemic has demonstrated the crucial role played by ICT in the education sector, specifically in the support of remote learning. Recent studies like [6] revealed that during school closures, teachers faced challenges related to student engagement and students' lack of Internet access, and that the challenges were more prominent in

high-poverty schools and rural areas. Approximately 43% of teachers reported concerns related to communication with students and student participation, while teachers in higher-poverty schools were more likely to indicate that their students did not have Internet access at home. It is not difficult to imagine how the technologies presented in Sections 3 and 4 could help bridge such gaps. In Figure 3, NTN solutions, such as GEO/LEO satellites or UAVs, may serve as space/aerial access points providing internet access to schools as well as to students' houses located in remote or critical areas. As a matter of fact, UAVs and other aerial appliances have always played the role of Internet carriers in poorly connected areas, such as for disaster recovery. However, many studies also consider them as a means for bridging the digital divide in education [41]. A pilot experiment was conducted by Google through its Project Loon [42], which aims to bring connectivity to millions of people that are offline in rural New Zealand, or Facebook Aquila (https://engineering.fb.com/2018/06/27/connectivity/high-altitude-connectivity-the-next-chapter/). Moreover, within each school building, smart antennas and intelligent surfaces can enable dynamic resource allocation policies, so that the bandwidth is allocated to different indoor locations/rooms in a fine-grained way based on the activities being performed. In the same way, service-fruition 6G technologies can result in increased effectiveness of remote/in-presence teaching, higher student engagement, and customized experiences based on the students' needs. Affective computing constitutes one of the most active research topics in education via the integration of visual and textual channels according to the survey in [31]. It can also help in detecting abnormal situations with the scholars at early stages, such as Attention Deficit and Hyperactivity Disorder [43] that otherwise would cause students to easily detach from the learning environment. Educational scenarios could also take advantage of XR. In particular, seminal XR-based solutions have been implied in education, mostly for medical purposes [44], with successful outcomes. However, it is in pedagogical use that XR technologies arguably unleash their most significant outcome: their ability to enhance active and experiential learning [45]. With such technology at hand, even scholars that are not physically collocated can share experiences and collaboration. This can transform the way educational content is delivered to students, encouraging creativity and bringing abstract concepts to life.

Figure 4 shows a comparison of the two analyzed use cases in terms of importance level of the technologies discussed in Sections 3 and 4 for their realization.

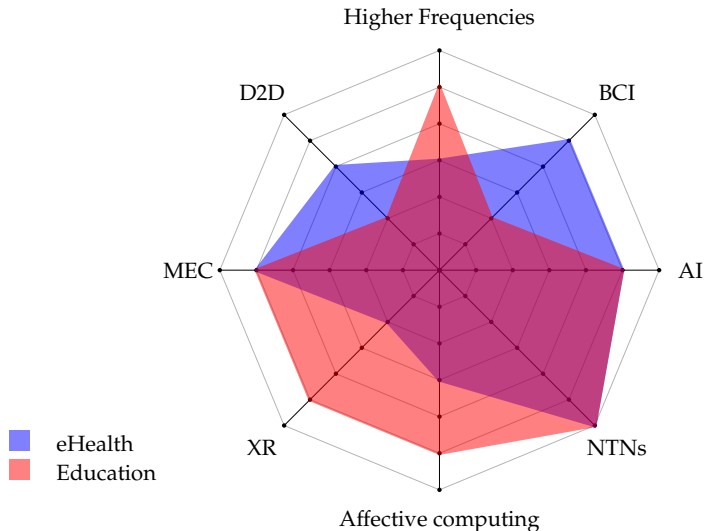

**Figure 4.** Relevance of the 6G technologies on the two use cases.

## 7. Discussion and Conclusions

The 6G is foreseen to support ground-breaking requirements. Top-notch technologies will have to be exploited by the future generation networks to foster high-performance

management of the new expected use cases, including those that arose during the health emergency caused by COVID-19. Due to the lockdowns imposed by the pandemic, work organization, education, and other daily-life habits have been *contaminated* by technology in ways that cannot be defined as temporary. We could state that the acceleration experienced by the digitization process, in the last year, represents the silver lining of COVID-19. Indeed, the pandemic has shifted the attention of worldwide researchers and government institutions to the importance of technological progress and, consequently, even to the problem of the digital divide, which affects people who have been excluded from the progress achieved thanks to the stimulus for improvement favored by the health emergency and investments in the ICT sector. At the same time, although digital transformation may represent a lifeline in many cases, opinions exist for which it could represent an additional cause of exclusion for the so-called *have-nots*. Considering the evolution of wireless mobile networks, 6G could exacerbate the digital divide, given that already the 5G standard has been unevenly distributed in different countries of the world in the first phase of release, thus leading to the creation of a further gap between the areas that receive state-of-the-art network coverage and those that do not. As we also stated in Section 2, there are contradictory opinions according to which technological evolution could solve or, on the contrary, aggravate the phenomenon of the digital divide; however, COVID-19 has put the spotlight on the problems relating to the profound inequalities in access to information and communication technologies, thus triggering various world institutions to conduct studies on the impact of the pandemic also on the economic field. For example, the report released by United Nations University (UNU) and United Nations Institute for Training and Research (UNITAR) highlights the exacerbation of the digital divide caused by COVID-19, reporting a 30% fall in sales of electronic and electrical equipment in the low- and middle-income countries, and only 5% in high-income countries [46]. This argues that the pandemic has hurt the worldwide economy but the poorest countries harder, testifying to the fact that at the root of the problem of the digital divide there are also significant economic disparities, the solution of which is outside the scope of this paper. In fact, this work aims to illustrate the evolution of the digital divide phenomenon and place it in the context of the future 6G, providing an overview of technologies and applications that telecommunications companies could exploit to help in bridging this gap, which has existed since technology began to spread among the people. In the vein of the decentralization trend described in Section 3, also 6G business models are expected to be decentralized to support the new use cases. The birth of the telecom virtual operators (MVNO) or the network sharing business model are concrete examples of how business dynamics are increasingly moving away from the classic model of centralization of resources at the mobile network operator (MNO) [47]. This can bring benefits in terms of extension of network coverage and enhancement of the provided services. For example, the newly-introduced micro operators can manage the requests of specific target users by offering higher-quality localized connectivity customized to the consumer's needs.

**Author Contributions:** Conceptualization, C.S. and S.P.; writing—original draft preparation, C.S., S.P., F.M., and M.D.F.; writing—review and editing, C.S., S.P., F.M., M.D.F., and G.A. All authors have read and agreed to the published version of the manuscript.

**Funding:** This research received no external funding

**Informed Consent Statement:** Not applicable

**Data Availability Statement:** Not applicable

**Conflicts of Interest:** The authors declare no conflict of interest.

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
