# Peer review of "6G to Take the Digital Divide by Storm: Key Technologies and Trends to Bridge the Gap"

_futureinternet, doi:10.3390/fi14060189_

Round 1
Reviewer 1 Report
The topic of this manuscript is important. As the authors themselves argue COVID-19 showed us that there is an urgency to bridge digital divide.
Lack of network infrastructure is naturally the key element to address this issue. Also, service-fruition divide plays a role that contains e.g. digital illiteracy, age, handicaps and other conditions. I believe that taking service-fruition into account gives more holistic picture of the current problems concerning digital divide.
This article presents how sixth generation (6G) systems and key technologies can bridge service-fruition divide, and authors introduce two uses cases, namely eHealth and education. They also consider socio-economical impacts of future 6G solutions
The article is well written, and it flows logically. Findings are focused on a technological level, and what possibilities these new emerging areas could bring to the table.
I would have hoped to see some discussion of how we actually make sure that these new technologies are used to fix problems and not to make more profit. I see the potential, but cynical side of me believes that these technologies will also increase digital divide. It’s only natural that companies will be mostly focused on the needs of their heavy users. It is more lucrative to keep current customer happy than to get a new one.
The use cases of eHealth and Education are quite short, and the findings are somewhat shallow. Both sections introduce valuable examples, but there would be much more to uncover. For example, special care consultation between doctors is important as well. In education there probably is a huge potential when using AI. I understand that journals have their page limitations, and this might be the cause for not going deeper into either of these cases. Now they serve well as examples of 6G possibilities.
Overall, this study is highly interesting. Service-fruition viewpoint is valuable, and I encourage the authors to continue this topic.
Reviewer 2 Report
The authors of this paper study the way that future 6th generation systems can remedy actual limitations in the realization of a true digital world. To this end, they introduce the key technologies for bridging the digital gap and particularly show how they can work in two uses cases of particular importance, namely eHealth and education, where digital inequalities have
been dramatically augmented by the pandemic.
The contribution of this paper can be summarized in the following three directions:
a) the authors discuss the evolution of the digital divide concept and the challenges that can be addressed by the technological development;
b) the authors discuss how the future 6G network is expected to overcome both the issues, by identifying trending technologies that should be further
developed and be part of the upcoming specifications. In addition, the authors study the impact of Artificial Intelligence and big-data collection and analytics via Machine Learning techniques
c) the authors present selected use cases (e.g., eHealth and education).
This a very interesting paper, whose subject is very well presented and will help the readers understand this significant research area.
The literature review provided by the authors is adequate and the whole presentation of the subject is quite good. No major syntax/grammatical errors have been found.
Based on the above, the reviewer believes that the paper can be accepted for publication.
Reviewer 3 Report
This manuscripts presents an overview of some 6G related technologies that could help bridge the gap created by a digital divide in society. Although there are some English errors here and there, the manuscript is generally clear and well written. A revision with an native-English speaker is suggested. Some other comments:
1. This manuscript cannot be considered as a research article, as it only makes a review of previously published articles. It is strongly suggested to change to the category of Review paper. Otherwise, the manuscript should be rejected.
2. The main concern with this paper is the very short number of references it reviews. Most subsections (as 3.1, 3.3, 3.4, 4.1, 4.2, 4.3), in which key technologies are discussed, cite only one reference. With such a limited number of references, it is not possible to make a strong case for what the authors are arguing for. It is strongly suggested that the authors find and include a larger number of works to include in their discussion.
3. There is a poor discussion on how the mentioned technologies and applications could help bridge the digital divide gap. Most of the technologies and applications are discussed in a very brief sub-section with a compelling argument. Most of the discussion feels like it is only the author's opinion, and not an accurate description of current state-of-the-art technological developments.
Reviewer 4 Report
This well-written manuscript described the prospective of the 6G technologies and their applications. One concern about this paper:
A very popular reason of the inability to access the digital world is economic, how the 6G technologies can bridge this gap?
Round 2
Reviewer 3 Report
Thanks to the authors for providing an answer to this reviewer's comments. More references are included and the discussion has been extended. The author's recognize the need of changing the manuscript's category from "Research Article" to "Review Article." No further comments.